# Multiplexed Fluorescence Plate Reader In Situ Protein Expression Assay in Apoptotic HepG2 Cells

**DOI:** 10.3390/ijms24076564

**Published:** 2023-03-31

**Authors:** Rita Jakabfi-Csepregi, Gábor L. Kovács, Péter Kaltenecker, Tamás Kőszegi

**Affiliations:** 1Department of Laboratory Medicine, Medical School, University of Pécs, 7624 Pécs, Hungary; ritacsepregi93@gmail.com (R.J.-C.);; 2Szentágothai Research Center, University of Pécs, 7624 Pécs, Hungary; 3Hungarian National Laboratory on Reproduction, University of Pécs, 7624 Pécs, Hungary

**Keywords:** multiplexed assay, signal protein expression, HepG2 cells, apoptosis, dose–response analysis, Ampliflu Red, fluorescence, plate reader

## Abstract

Instead of Western blot being considered as a gold standard for intracellular protein expression assays, we developed a novel multiplexed high throughput (180 tests/day) in situ manual protein expression method directly in 96-well plates using 25,000–100,000 cells/well after formaldehyde fixation and Triton X 100 permeabilization. HepG2 cells were treated with ochratoxin A (OTA) and staurosporine (STP) to induce apoptosis. Antioxidant and apoptotic cell signaling protein expression were studied by various rabbit primary antibodies and HRP labeled secondary antibodies. The HRP labeled immune complexes were developed by H_2_O_2_/Ampliflu Red fluorogenic reagent and measured in a plate reader. Our assay can simultaneously quantify 22 protein antigens in one plate with 4 technical replicates with an interassay imprecision of <10% CV. The fluorescence signals are referred to total intracellular protein contents in the wells and given as fluorescence/protein ratio FPR, expressed as % of the controls (FPR %). OTA caused a dose–response increase (*p* < 0.05–*p* < 0.001) in SOD2, CAT, ALB, CASP3,7,9, BCL2, BAX, Nf-kB, phospho-Erk1/2/Erk1/2, phospho-Akt/Akt, phospho-p38/p38, and phospho-PPARg/PPARg levels while phospho-AMPK/AMPK ratios decreased (*p* < 0.05–*p* < 0.001). On the contrary, STP induced a dose–response decrease (*p* < 0.05–*p* < 0.001) in CASP3,7,9, BAX, BCL2, Nf-kB and phospho-Erk1/2/Erk1/2 expression while B-ACT, phospho-Akt/Akt, phospho-p38/p38 and phospho-PPARg/PPARg ratios increased.

## 1. Introduction

A better knowledge of intracellular cell signaling pathways is essential for understanding major life processes at the molecular level. To obtain quantitative data on signal proteins two types of approaches are usually applied: i, gene expression (mRNA) studies by qPCR methods; ii, protein expression analyses frequently performed by Western blot techniques but single cell methodologies and/or microscopic examinations are also available [1,2,3,4]. However, the question if mRNA levels are proportional with the extent of their coded proteins’ translational rate is still under extensive investigation [5]. In general, the mRNA contents are proportional with their corresponding protein products but the transcript levels do not necessarily predict the amount of translated proteins [6]. Therefore, it is of utmost importance to quantify individual intracellular proteins in tissue culture models.

There are two main types of gold standards for quantification of individual proteins in complex cellular samples: ELISA and Western blot protocols. Generally, the ELISA technique (if commercially available) is easier and faster to perform because the procedure lasts for a few hours only and it can be automated as well. However, calibrators and controls of the protein in question are necessary. In addition, ELISA can be done using a 96-well precoated plate with low amounts of protein samples due to the method’s high sensitivity (ng/mL or less). However, in the ELISA technique there is no internal control for the studied samples just their total protein contents. For antigens with no calibrators the usage of ELISA technique is questionable. In contrast, Western blot maybe used without purified standards because of the applied internal load controls. The main advantage of Western blot is that the previous separation step can provide extra information on the molecular weight of the target protein. It can exclude non-specific binding due to the different molecular masses of the analytes if multiple bands are detected on the membrane. In this way, it provides information on the purity of the sample. This contrasts with other techniques, where non-specific binding can significantly compromise the validity of the results [7,8]. However, Western blotting (especially the manual ones) may carry several errors from sample preparation to signal processing. A major concern is signal saturation or high background and the problems of densitometry of the images of the bands (uneven staining, distortion, high background, etc.). It cannot be excluded that the signal of the applied loading control is not stable because the treating agent may influence the expression of the housekeeping protein [9]. Although the Western blot test is labor intensive and time consuming as it requires manual intervention at various stages of the process, this technique is often preferred to ELISA. One of the major limiting factors in the case of Western blot is the transfer or blotting step [10], which is also known to be highly variable depending on the membrane type and porosity that influences the transfer efficiency of proteins with different molecular weights (between 60% and 100%) for some hydrophobic membranes, thus making quantification difficult. This is a particularly problematic step for large molecular weight proteins that can be difficult to transfer effectively [7,11]. In addition, quite often a large amount of sample (5–15 µg) would be needed for adequate results [12], which can jeopardize the analysis when the amount of sample is limited [13]. Furthermore, high volumes of buffers and antibodies are also required, which makes the procedure expensive. Finally, the analysis is often limited to a single target protein [14]. The detection of multiplex target proteins requires the removal of bound antibodies (a process well known as stripping) before re-probing. This increases the analysis time and can cause a significant loss of target proteins [15].

To summarize, Western blotting for single protein quantification seems to be more difficult than the molecular biology approach (qPCR) because intracellular quantitative protein expression usually requires cell lysis with protease inhibitors, SDS-PAGE followed by various types of blotting procedures, blocking and immune reaction usually with chemiluminescence detection (ECL) [16]. Additionally, instead of highly specific primers, we need primary and enzyme-labeled secondary antibodies with various specificities and a successful method of detection of the immune complexes which is quite often relative because of the lack of calibrators. To overcome the described difficulties in the manual methods, several partially or fully automated techniques have been introduced [17]. However, even the above automated methods require cell lysis procedures, and an internal standard (load control) that is thought to be not affected by the various treatments in the experiments. The light signal of the immune complex (protein band) developed by fluorescence or ECL measurements should be referred to that of the internal load control (beta-actin or GAPDH) giving only a ratio to compare the protein expression after the different treatments [18]. Although there are multiplexed assays with simultaneous detection of several protein antigens, these methods require highly sophisticated instrumentation and expensive consumables as well [19,20]. An additional shortage of manual Western blotting is the low number of independent replicates that makes statistical evaluation a bit questionable.

Therefore, our aim was to establish and validate a multiplexed fluorescence in situ intracellular protein assay that does not require the critical steps needed for performing Western blot analyses (cell lysis, denaturation, electrophoresis and detection of the immune complexes referred to as load control proteins). Our simplified technique was based on the fixation and permeabilization of tissue cultures directly in 96-well culture plates followed by multiplexed immune reactions and signal processing. All subsequent procedures were performed on the same culture plate.

To test our novel in situ protein expression system, we used apoptosis models where characteristic signal protein expression changes occur during the apoptotic process. Another reason to choose apoptotic protein quantification is that the wide-spread and accepted fluorescence flow cytometric methods for characterization of apoptotic cells require expensive instrumentation and various fluorescence labels (Annexin V, propidium iodide, 7-AAD, etc.) that usually give just the relative amounts of apoptotic cells in a percentage distribution. On the other hand, flow cytometry has inevitable advantages that it might discriminate the various cell populations (unaffected, early apoptotic, late apoptotic and necrotic) but it does not give information on the key protein changes during the process unless the assay is combined with protein and viability labels as well to provide multiparametric data [21].

In our model, we chose HepG2 cell cultures that are well characterized and can express human albumin as well. In a 2D environment, the albumin expression is quite low compared to that in long-term 3D Matrigel-based conditions [22]. To induce apoptosis, we chose two various types of chemicals with different mode of actions: ochratoxin A mycotoxin (OTA) and staurosporine (STP). Both agents cause apoptosis and especially OTA in a dose-dependent way necrosis as well [23,24,25]. The mode of action of the two compounds is not identical. Both OTA and STP causes energy depletion but in a different way. OTA inhibits protein expression, induces ROS formation and can cause arrest of the cell cycle [26,27,28]. The primary action of STP is thought to be the inhibition of mitochondrial kinases by competition with ATP availability with a consecutive release of cytochrome c and a subsequent activation of caspases-9 and -3 by an apoptosis-like mechanism [29,30,31]. Because of the complex action of OTA, in our experiments, the antioxidant scavenger enzymes and apoptosis markers were investigated together. However, in the regulation of intracellular processes, phosphorylation is of key importance both in physiological and in pathological conditions (e.g., tumor cell propagation) [32]. Therefore, we tested our system to see if phosphoproteins could also be quantified by our novel assay. In the case of STP, a major focus was performed on the estimation of phosphorylated signal proteins due to its major action of protein kinase inhibition. In our work, we validated the in situ multiple protein quantification technique and tried to obtain information on the various mode of actions on HepG2 cells between OTA and STP treatments. The following intracellular proteins were analyzed: superoxide dismutase 2 (SOD2), catalase (CAT), caspases 3,7,9 (CASP3, CASP7, CASP9), B-cell lymphoma-2 (BCL2), BCL2-associated X protein (BAX), nuclear factor kappa-light-chain-enhancer (Nf-kB), glyceraldehyde-3-phosphate dehydrogenase (GAPDH), human albumin (ALB), β actin (B-ACT), p38 mitogen-activated protein kinases (p38), protein kinase B (Akt), peroxisome proliferator- activated receptor gamma (PPARg), 5′ AMP-activated protein kinase (AMPK), and extracellular signal-regulated protein kinases (Erk 1/2). In the case of some cell signaling proteins, the phosphorylated forms were also measured: p-Akt, p-p38, p-PPAR-g, p-Erk 1/2, and p-AMPK.

## 2. Results

### 2.1. Validation of the In Situ Multiplex Protein Expression Assay

#### 2.1.1. Antibody Dilution Optimization

First, we had to optimize the applied dilutions of the primary and secondary antibodies (ABs) with a checkerboard assay. We found that in the case of most antigens, an 800-fold dilution could be applied, 2000-fold for ALB and B-ACT, and for GAPDH, 5000-fold primary antibody dilutions were used. The secondary antibody proved to be the best at 4000-fold dilutions for each immune reaction. The blank (without any AB and with secondary AB only) showed identical fluorescence signals that were negligible in most cases when compared with those obtained for the immune sandwiches. Typically, the blank values were less than 9% when compared with the signal of the whole immune complexes with 3 exceptions (Nf-kB, Akt and p-Akt), Table 1.

Because the blank signals were practically uniform, we did not subtract the blank values in the calculations. Our primary antibodies were all of rabbit origin; therefore, we could use the same HRP labeled secondary antibody. The fixation depleted all intracellular peroxidase activity that was proven by pre-treatment of the samples with 3% H_2_O_2_ exposure without any significant impact on the immune complex formation and on the detected fluorescence signal (blanks and immune sandwiches as well) compared with the data without H_2_O_2_ pre-treatment (not shown here).

#### 2.1.2. Cell Number Optimization

The number of treated cells vs. the concentration of the applied treating agents may strongly determine the results of the experiments. Therefore, we had to establish an optimal cell number range plated into the wells that uniformly responses to the exposure, independently of the treating agent’s concentrations. OTA was tested by GAPDH housekeeping protein expression at zero and 10 µM levels, the latter having definite effects on viability parameters as it was previously shown [25]. We obtained an almost uniform response of HepG2 cells from 25,000 to 100,000 cells/well plated. However, less than 25,000 cells in the wells showed a much higher response (FPR) indicating that OTA could exert a stronger effect on the cells (relatively higher OTA/unit cell number), Figure 1.

We would like to emphasize that the initial cell numbers plated increased a bit after 24 h of preculturing which slightly increased further during OTA treatments (another 24 h). To be able to compare the control and exposed groups, the control cells were incubated under identical conditions. Our data showed that within the range of 25,000 to 100,000 plated cell numbers the protein contents of OTA treated cells did not differ significantly from those of the controls as tested by Student’s *t* test (*p* > 0.07–*p* = 0.12)); however, at 25,000 cells, the FPR values slightly increased.

#### 2.1.3. Intraassay and Interassay Imprecision Studies

We used GAPDH expression for the intraassay imprecision estimation that is considered to be a housekeeping protein without any changes after various treatments. For intraassay studies, 84 parallels within one plate without any treatment showed an average FPR of 100 ± 9.08 SD. For the interassay experiments, beside GAPDH, we also measured the protein expression of B-ACT in control cells and in cells treated with 10 µM OTA. In general, the coefficient of variation of 4–8 independent experiments did not exceed 10% for all studied proteins. Our imprecision data are summarized in Table 2.

### 2.2. Ochratoxin A Treatments

We used an OTA concentration range that covers minimal changes to significant effects both in viability and in apoptosis rates. Previously, it was shown that OTA is considered to cause ATP depletion with oxidative stress and disturbed protein synthesis causing apoptosis and necrosis at the same time [23,25]. We studied antioxidant enzyme expressions together with key signal proteins related to apoptosis. Additionally, the so-called housekeeping protein expressions were analyzed including human albumin (ALB) which was thought to be synthesized by HepG2 cells. In our experiments (6 independent measurements with 4 technical replicates for each antigen), we could observe a dose–response increase in both the antioxidant enzymes and in the main apoptosis markers. However, GAPDH and B-ACT levels were unchanged, but ALB expression showed a marked increase (Figure 2a,b).

In the next series of experiments, signal protein expressions and their phosphorylated forms were studied together, all compared with the proteins’ FPRs of the untreated controls in %. A significant dose–response increase was found after OTA exposures in the case of PPARg, Erk 1/2 and AMPK while the phosphorylated p-p38, p-PPARg and p-Erk 1/2 also showed a dose–response type elevation (Figure 3a,b). When the ratio of phosphorylated/total signal proteins was calculated, the p-Akt/Akt, p-p38/p38, p-PPARg/PPARg and p-Erk1/2/Erk1/2 ratios increased parallel with OTA concentrations while the p-AMPK/AMPK levels decreased accordingly (Figure 4).

### 2.3. Staurosporine Treatments

Staurosporine is thought to be an overall apoptosis inducer in practically all types of mammalian cells. Its major mode of action is the inhibition of protein kinases through interaction with the kinases at their ATP binding sites with a strong and non-selective way [33]. One important action of staurosporine in apoptosis induction is the activation of the caspase-9 mediated proteolytic pathway with consecutive activation of further caspases by limited proteolysis [34]. It is generally believed that another mechanism of apoptosis induction is the mitochondrial attack resulting in the release of cytochrome c with further ATP depletion [35].

A major technical advantage in cellular model systems regarding the effects of STP exposure is the widely accepted short exposure time (6 h) that speeds up the whole cellular labeling procedure.

We measured the main apoptosis markers as previously shown but without SOD2 and CAT detection (Figure 5a,b).

Except ALB and GAPDH, we obtained a dose-dependent decrease in the apoptosis marker levels however, quite unexpectedly B-ACT expression increased. Some native and phosphorylated signal proteins were also quantified. Staurosporine in the range of 0.5–1.5 µM caused a dose-dependent decrease in Akt, p38, Erk 1/2 and AMPK levels, while the phosphorylation pattern was diverse. Phosphorylation generally was less in all cases of the studied proteins but for p-p38 and p-PPARg the decrease in native proteins showed a parallel tendency to that of the phosphorylated forms (Figure 6a,b).

When calculating the phospho/native protein ratios the highest dose-dependent increase was found for p-Akt/Akt but the p-Erk 1/2/Erk1/2 ratios decreased dramatically. However, the other ratios did change only mildly and in a variable way with accidental significance levels (Figure 7).

## 3. Discussion

We established a multiplexed simple fluorescence in situ intracellular protein expression assay adapted to a multiplate reader. All treatments and manipulations could be performed inside the wells of the 96-well culture plates. The formaldehyde fixed and detergent permeabilized cells remained attached to the bottom of the wells enabling all washing and labeling steps to be easy to perform. The primary AB dilutions (from 800-fold to 5000-fold) were very economical when compared with traditional Western blot technology where much more cells (cultured in 6-well plates) and larger volumes of Abs are needed. We used primary Abs of rabbit origin but if the primary Abs within one plate are not uniform (raised in other animals as well), it does not mean any difficulties because in the different wells, different anti-animal HRP labeled secondary ABs can be used. Our method eliminates the need of cell lysis, denaturation, SDS-PAGE, transfer, and chemiluminescence development/densitometry of the signals together with internal (load) controls. The classical Western blot assays are time consuming and low throughput ones with many potential errors during the whole procedure. Moreover, most of the applied manual Western blot methods can quantify only a few antigens within one run. Another shortage of the Western blot analysis is the usually low number of independent experiments and/or technical replicates required for statistical calculations. In our work, we addressed a general experimental setup problem often not mentioned in the literature namely, the concentration of treating compounds/treated cell number. If a quasi-uniform response range regarding compound levels/cell number is not achieved, then the data found in the literature/experiments cannot be interpreted properly. Therefore, we optimized the initial cell number range where there was practically no significant change within 25,000–100,000 cells/well (Figure 1).

Our robust multiplexed assay could be completed manually within 6–7 h when blocking was performed at 37 °C. The signal processing was performed by spectroscopic precise quantification instead of the more cumbersome densitometry. One skilled technician could handle 2 plates simultaneously with 30 min shifts in the manipulations that results in theoretically about 180 data/day with maximum 22 antigens/plate detected, each of them with 4 technical replicates. In most cases, the widely used B-ACT and GAPDH expression did not change. However, in the STP exposure experiments B-ACT levels increased in a dose-dependent way (Figure 5a). This means that each load control in Western blot analyses should always be carefully checked for its constant translation rate.

Although there are numerous publications on the mode of action of OTA and STP, the complexity of the induced intracellular processes still raises open questions. The diversity of literature data regarding OTA toxicity strongly depends on the different experimental conditions (cell lines, exposure time, FBS present or not in the media, toxin concentration vs. treated cell number, etc.). OTA shows a strong binding characteristic to albumins of various species [36,37]; therefore, FBS containing culture media require relatively high (in the µM range) toxin levels and long-term incubation. The uptake of OTA is mediated by transporters (e.g., organic anion transporter, OAT) [38]. Besides the well-known OATs, the OTA transporters may consist of several families of multi-specific organic anion transporters, organic anion-transporting polypeptides (OATPs), oligopeptide transporters (PEPTs), and ATP-binding cassette (ABC) transporters (MRP2 and BCRP) [39]. Intracellular OTA exerts its toxic effects through various mechanisms: ATP depletion, inhibition of protein synthesis, ROS formation, direct DNA damage and interfering with cell signaling mechanisms [39]. It was shown that in pluripotent hESC cell cultures OTA exposure causes decreased viability and proliferation ability, and the increased ROS production induces the overexpression of SOD2 and CAT antioxidant enzymes [40]. In our experiments, the increase in the antioxidant enzymes in HepG2 cells differed significantly from that of the controls above 5 µM of OTA present (Figure 2a). The fine resolution of our method is supported by the significant differences found between 7.5 and 10 µM OTA exposures (*p* < 0.001). As a general consequence of ochratoxin treatment, the cells undergo apoptosis and at high toxin levels necrosis as well with a cell cycle arrest [25,28,40]. A major route in the apoptotic process lies in the activation of caspases [23,41]. Caspases are also involved in inflammation, cell cycle and cell differentiation processes. In apoptosis, the initiator seems to be CASP9, further activating CASP3,6,7 and other signaling pathways [42]. In our experiments, OTA caused a dose-dependent increase in the expression of CASP3,7 and 9 together with BCL2, BAX and Nf-kB (Figure 2b). BCL2 proteins mediate several processes from development, homeostasis, autophagy, and innate to adaptive immune responses. The unbalanced expression of BCL2 is often related to various diseases including cancer. In mammalian tissues, there are various BCL2-related proteins (with pro-survival functions as BCL2, BCL-xL, BCL-w, Mcl-1, A1, BCl-B, and with proapoptotic function Bax, Bak, and Bok). Moreover, there are other related proteins (Bim, Bad, Bmf, Bid, Bik, Noxa, Puma, and Hrk, respectively) [43,44]. In general, BCL2 is believed to exert an anti-apoptotic effect; therefore, in apoptosis its transcription level in most cases is downregulated. The balance between the pro-survival molecules of the BCL2 family (BCL2, BCL-XL and MCL1) and the two pro-apoptotic proteins (BIM, PUMA and BID) will determine if the death-related signal proteins BAX and BAK are activated causing the perforation of the mitochondrial outer membrane that triggers the proteolytic cascade to force the cells to undergo apoptosis [45].

In our study, we observed an unexpected dose-dependent increase in BCL2 expression besides the expected increase in BAX, Nf-kB [45,46] and to a less extent that of albumin. These findings may be explained by the combined action of OTA inducing ROS and apoptosis/necrosis as well, but we must note that in general, necrosis may also occur at above 10 µM OTA exposure under our experimental conditions [25]. The increased ALB expression may be attributed to its antioxidant features detected in human serum [47]. Phosphoproteins might have several roles, depending on the phosphorylation site and the protein function as well but phosphorylation can usually be considered as an activation process [48]. OTA did not induce increased Akt and p38 levels but PPARg, Erk 1/2 and AMPK showed a dose-dependent elevation. On the other hand, the phosphorylation pattern was not uniform: both phospho-Akt, p-p38 and p-PPARg increased while p-AMPK was decreasing with an elevation in OTA levels. Our data regarding the Akt pathway are at least partially in agreement with the results of Özcan et al. [49].

Staurosporine is a potent apoptosis inducer, but its mechanism is largely unknown and is not identical to that of OTA. However, the major action of STP is thought to be the inhibition of various protein kinases (PKs) with weak specificity [50]. Apoptosis is considered to undergo by extrinsic and intrinsic mechanisms which differ at the beginning but finally they may merge into a common pathway [51]. During the intrinsic signaling mechanisms the mitochondrial membrane is affected with the subsequent release of cytochrome c, stimulating/inhibiting the pro-apoptotic factors BAK, Bad, and BAX and the anti-apoptotic factors BCL2 and BCL-xl. The result is the activation of CASP9 with a consequent proteolytic cascade where CASP3 induces apoptosis [30]. Although, there are several other signal molecules of significance in the apoptotic process, we wished to see if there are any differences in the protein expression related to two different types of programmed cell death inducers (OTA vs. STP). The highest degree of downregulation was observed for CASP7, BCL2 and Nf-kB while the other caspases decreased only at the highest STP concentrations. BAX also decreased but did not show dose-dependency and ALB and GAPDH expression did not change. As for the phosphoproteins, despite the significant downregulation of Akt, the ratio of p-Akt/Akt dramatically increased. Opposite to Akt the p-Erk 1/2/Erk 1/2 decreased to about 50%. In all other cases, the phospho/native protein levels changed proportionally without any significant alterations (Figure 5, Figure 6 and Figure 7).

Due to its strong apoptosis induction feature, it might be plausible to use staurosporine as an anticancer drug, but its toxicity is too high. However, some chemical derivatives of staurosporine may be tested as potential anticancer drug candidates [52]. To decipher the actions of STP on cell cultures beside the expression of apoptosis-related signal proteins, their mRNA expression should also be studied.

In summary, our novel assay was suitable for the sensitive and reproducible quantitative determination of some important cell signal protein expression changes in a HepG2 model system. We could discriminate between the mode of action of the two studied toxic compounds OTA and staurosporine. Our high throughput method was also suitable for the measurement of phosphoprotein expression. The low CV % at a high independent and technical replicate number of measuring the expression rate of various signal proteins makes statistical analyses more precise compared to the classical Western blot technique where conclusions are usually drawn from 3–5 repeats. A shortage of our assay is similar to most of the classical Western blot techniques namely, we could not give an absolute amount of the studied proteins but their relative amounts as a fluorescence signal/total protein ratio (FPR).

In the future, we would like to test our system in cell suspensions as well. When dealing with suspensions a major drawback of the method is that all manipulations should be performed in Eppendorf tubes and repeated centrifugation steps are necessary. We also plan to analyze the gene expression of the studied signal proteins by qPCR method.

## 4. Materials and Methods

### 4.1. Cell Cultures

HepG2 cell cultures (ATCC-HB-8065, ATCC, Manassas, VA, USA) were maintained in Dulbecco’s modified Eagle’s medium-high glucose (DMEM, 4500 mg/L), supplemented with 10% *v*/*v* fetal bovine serum (FBS), both from Biosera Europe Nuaille, France. The medium was completed with penicillin-streptomycin (100 µg/mL, Merck, Darmstadt Germany). Cells were pre-cultured in 25 cm^2^ flasks at 37 °C in a humidified 5% CO_2_ atmosphere incubator until 80% confluency and after trypsinization, in most of the experiments 10,000 cells/mL were added to the wells of 96-well culture plates (TPP, Merck, Darmstadt, Germany). Before treatments, the cells were further cultured in the plates for 24 h, to reach at least 25,000 cells/well.

### 4.2. Chemicals

Ochratoxin A (OTA) from Merck, Darmstadt Germany was dissolved in 96% *v*/*v* ethanol (Ph. Eur.) to obtain a 5 mM stock solution and was kept at 4 °C. Staurosporine (STP) was from Bio-Techne R&D Systems, Budapest Hungary. STP was dissolved in spectroscopic grade DMSO (Molar Chemicals, Budapest, Hungary) at 10 mM concentration and was kept at −20 °C. The fluorogenic substrate Ampliflu Red and the peroxide free Triton X 100 nonionic detergent were from Merck, Darmstadt Germany. Ampliflu was dissolved in spectroscopic grade DMSO at 5 mM concentration and kept at −20 °C. All other chemicals were of analytical grade. Phosphate-buffered saline (PBS with or without Ca, Mg), Tris buffered saline (TBS) and 0.05% *v*/*v* Tween 20-TBS (T-TBS) were prepared in the laboratory. For cell fixation, 3.5% *v*/*v* formaldehyde (Ph. Eur.) in PBS, for permeabilization of the cells 0.1% *v*/*v* Triton X 100-PBS were used. Before antibody treatments, the fixed and permeabilized cells were blocked using Superblock T20 (PBS) blocking buffer (Thermo Fisher Scientific, Waltham, MA, USA). Total protein content of the fixed cells was determined by using sulforhodamine B (SRB) staining (Thermo Fisher Scientific, Waltham, MA, USA).

### 4.3. Antibodies

All the primary antibodies were polyclonal, raised in rabbits. The following antibodies were used: SOD2, CAT, CASP3, CASP7, CASP9, BCL2, BAX, Nf-kB, ALB, B-ACT, p38, Akt, PPARg, AMPK, and Erk 1/2 which were from Thermo Fisher Scientific, Waltham USA. GAPDH was from Cell Signaling Technology, Danvers, MA, USA. In the case of some cell signaling proteins, the phosphorylated forms were also measured: p-Akt, p-p38, p-PPARg, p-Erk 1/2, and p-AMPK (antibodies from Thermo Fisher Scientific, Waltham USA). The secondary antibody was goat anti-rabbit IgG, HRP conjugated (Thermo Fisher Scientific, Waltham, MA, USA). The labeling protocol is described in detail in Section 4.5.

### 4.4. Treatment of HepG2 Cells

The precultured and plated cells were treated with OTA at 5, 7.5 and 10 µM final concentrations in DMEM for 24 h. In case of staurosporine, the precultured cells were exposed at 0.5, 1, and 1.5 µM levels in DMEM for 6 h. The solvent concentration was kept below 1% *v*/*v* during the treatments in 150 µL media.

### 4.5. In Situ Antigen Labeling Protocol

After the treatments the HepG2 cells were directly labeled in the wells of the culture plates. Both the fixation, permeabilization and the immune reaction were performed in the culture plate. The individual proteins were quantified by adding Ampliflu Red [9] fluorogenic substrate to the horseradish peroxidase (HRP) labeled immune complexes and the fluorescence signal was measured in a photon counting plate reader. Instead of load controls, the fluorescence intensities were referred to the total protein contents inside the wells and fluorescence/protein ratios (FPR) were calculated. The whole procedure took about 4–7 h (two plates/day by one skilled person) and the FPRs were referred to those of the control cells (expressed as %, taken the controls’ FPR to be 100%). In one plate, 22 different antigens could be tested, each of them in 4 technical parallels. The step-by-step procedure is described below.

Thorough washing 3 times with Ca, Mg containing PBS.Fixing with 150 µL of 3.5% formaldehyde-PBS at RT for 15 min.After 3 washing steps permeabilization with 150 µL of 0.1% Triton X 100-PBS at RT for 20 min.Blocking with 150 µL of Superblock T20 at 4 °C overnight, tightly capped. Alternatively, blocking could be performed at 37 °C for 1 h with agitation to reduce processing time.Removal of blocker by tapping and addition of blocker-diluted antibodies into the wells in 100 µL volume. Incubation for 2 h at 37 °C with 400 rpm in an orbital shaker. The antibody dilutions were 800-fold in most cases except for GAPDH (5000-fold dilution), B-ACT and ALB (2000-fold dilution).Washing 5 times with T-TBS (200 µL/well).Incubation with secondary antibody 4000-fold diluted with blocker for 1 h at 37 °C with 400 rpm in the orbital shaker (100 µL/well).Washing 2 times with T-TBS and once with TBS (200 µL/well).Detection of the immune complexes in the emptied wells by incubation for 1 h at RT with 150 µL/well Ampliflu Red/H_2_O_2_ reagent (50 mM K-phosphate buffer of pH 7.5 Ampliflu 16 µM, H_2_O_2_ 10 mM dissolved by 0.1% citric acid, final concentrations). The detection reagent should be prepared before usage and should be kept on ice protected from light.Measurement of fluorescence intensities at 540 nm/580 nm excitation/emission settings in a Perkin Elmer EnSpire multimode plate reader (Per-Form Hungaria Ltd. Budapest, Hungary).

For each studied intracellular protein, blank samples were applied containing only the secondary antibody or just the blocker. In a 96-well plate, 11 different antigens with their appropriate blanks using 4 technical replicates were simultaneously measured. The flow chart of the method is seen in Figure 8.

### 4.6. Intracellular Protein Measurement and Data Calculation

The emptied wells were filled with 50 µL of 0.04% sulforhodamine B (SRB) solution in 1% *v*/*v* acetic acid and incubated for 1 h at RT (a modified method of Vichai V, Kirtikara K [53] omitting trichloroacetic fixation). Then, the wells were washed 4 times with 200 µL of 1% acetic acid and were completely dried. The bound dye was redissolved by treatment with 200 µL/well of 10 mM TRIS-HCl buffer (pH 10.5) shaking the plates for 2 min at 400 rpm. The absorbance of the samples was measured at 565 nm in the plate reader. All procedures from cell culturing to protein determination were performed in the same culture plates.

Protein expression referred to that of the untreated controls was calculated by dividing the fluorescence signal in cps by the absorbance of the corresponding wells (fluorescence/protein ratio, FPR). The data were compared to the FPR of the controls and expressed as percentage (%) of the controls taken as 100%. In this way, all varying fluorescence and protein data could be interpreted in a uniform way in our multiplexed assay.

### 4.7. Validation of the Protein Expression Assay

#### 4.7.1. Cell Number Optimization and Imprecision Studies

To obtain the ideal cell number for further experiments, GAPDH expression was studied in controls and in 10 µM OTA-treated cells at various cell seeding concentrations in the range of 10,000–100,000 cells/well, using 4 technical replicates for each cell number. Both FPR data of the controls and those of the OTA-treated samples were calculated, the latter being referred to the control FPR taken as 100%.

Intraassay imprecision for GAPDH expression was obtained by measurement of the FPR values in one plate at 30,000 cells/well without any treatment (N = 84).

Interassay imprecisions of GAPDH and B-ACT were calculated for both untreated and 10 µM OTA exposed HepG2 cells in 4–8 independent experiments with 4 technical replicates in each case (N = 16−32).

#### 4.7.2. Statistical Evaluation

One-way ANOVA test was used to compare FPR results of treated samples with those of controls. Imprecision studies were interpreted by calculating the mean ± SD and the coefficient of variation (CV%). An IBM SPSS Statistics (USA) software version 23.0 program was used where the significance level was set at *p* < 0.05.

## Figures and Tables

**Figure 1 ijms-24-06564-f001:**
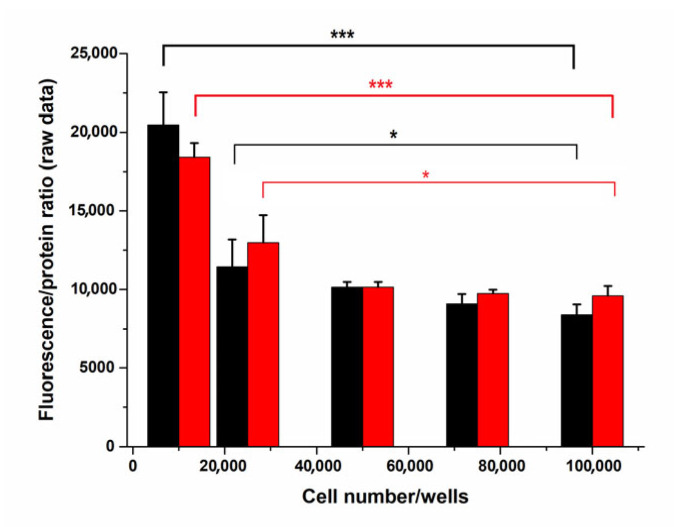
GAPDH expression in control and 10 µM OTA exposed HepG2 cells at various cell number/well concentrations (raw data as fluorescence/protein ratios). (N = 4). *: *p* < 0.05, ***: *p* < 0.001.

**Figure 2 ijms-24-06564-f002:**
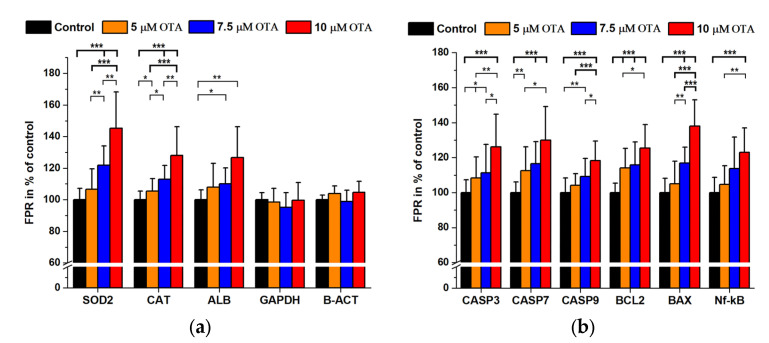
Antioxidant (**a**) and apoptosis-related signal protein expression (**b**) at various OTA concentrations expressed as FPR % of the controls (N = 6 × 4). *: *p* < 0.05, **: *p* < 0.01, ***: *p* < 0.001.

**Figure 3 ijms-24-06564-f003:**
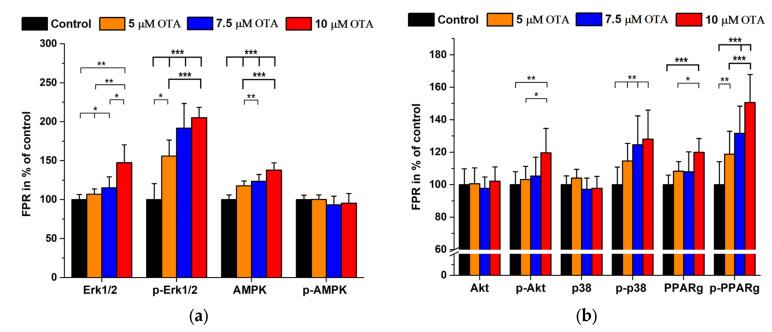
Signal protein and related phosphoprotein expression in OTA (0–10 µM) treated HepG2 cells. (N = 6 × 4). (**a**,**b**): data expressed as FPR in % of the controls. *: *p* < 0.05, **: *p* < 0.01, ***: *p* < 0.001.

**Figure 4 ijms-24-06564-f004:**
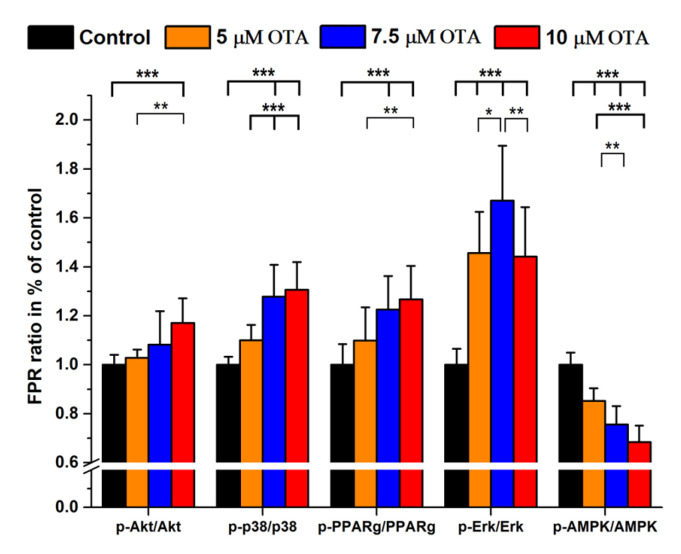
Phosphoprotein/protein expression in OTA (0–10 µM) treated HepG2 cells shown in FPR % ratios compared with those of controls. (N = 6 × 4). *: *p* < 0.05, **: *p* < 0.01, ***: *p* < 0.001.

**Figure 5 ijms-24-06564-f005:**
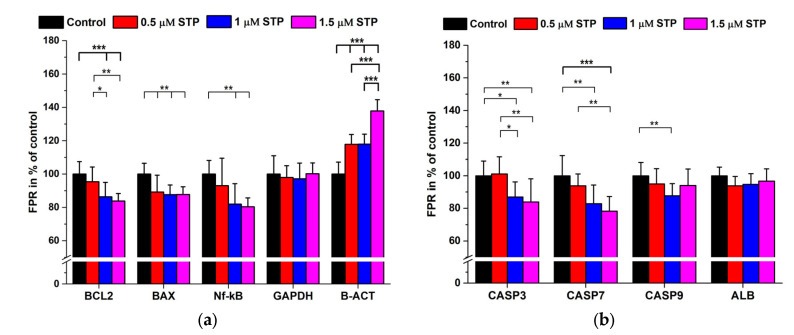
Apoptosis signal protein expression in STP (0–1.5 µM) treated HepG2 cells. (**a**,**b**): data are expressed as FPR in % of the controls (N = 6 × 4). *: *p* < 0.05, **: *p* < 0.01, ***: *p* < 0.001.

**Figure 6 ijms-24-06564-f006:**
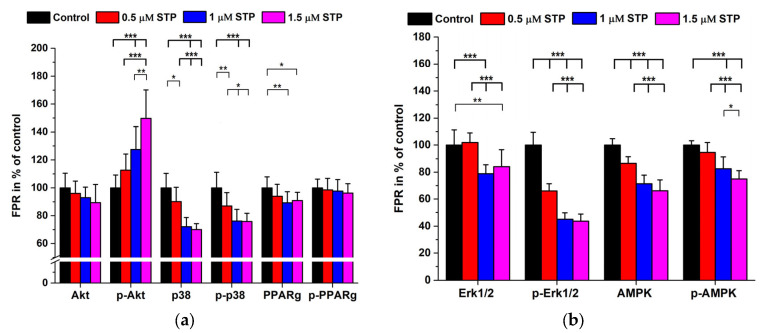
Signal protein and related phosphoprotein expression in STP (0–1.5 µM) treated HepG2 cells. (N = 6 × 4). (**a**,**b**): data expressed as FPR in % of the controls. *: *p* < 0.05, **: *p* < 0.01, ***: *p* < 0.001.

**Figure 7 ijms-24-06564-f007:**
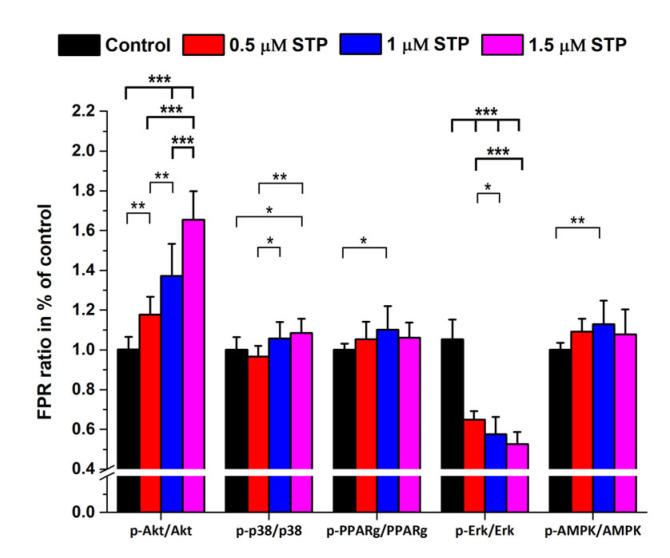
Phosphoprotein/protein expression in STP (0–1.5 µM) treated HepG2 cells expressed in FPR % ratios compared with those of controls. (N = 6 × 4). *: *p* < 0.05, **: *p* < 0.01, ***: *p* < 0.001.

**Figure 8 ijms-24-06564-f008:**
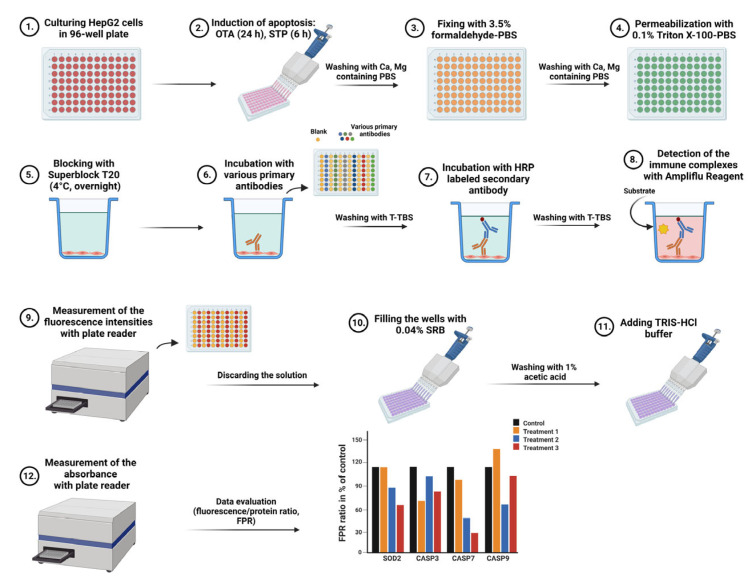
Flow chart of the multiplexed in situ protein expression method. “Created with BioRender.com”.

**Table 1 ijms-24-06564-t001:** Optimization of the primary and secondary ABs for the quantification of intracellular cell signaling proteins (N = 4 for each antigen). Background signals are given as % of the signal obtained for the immune complexes.

Antigen	2nd AB Only (Signal Mean ± SD) in Cps	Immune Complex (Signal Mean ± SD) in Cps	1st/2nd AB Dilutions	Background Signal in %
SOD2	482.0 ± 47.6	13,103.4 ± 1082.9	800/4000	3.7
CAT	587.0 ± 47.1	8403.1 ± 181.3	800/4000	7.0
Casp3	426.0 ± 40.1	11,738.1 ± 572.0	800/4000	3.6
Casp7	473.0 ± 45.4	11,033.3 ± 608.0	800/4000	4.3
Casp9	288.0 ± 22.7	13,338.1 ± 901.0	800/4000	2.2
BCL2	357.7 ± 39.9	4031.2 ± 91.4	800/4000	8.9
BAX	487.0 ± 62.6	7535.6 ± 421.0	800/4000	6.5
Nf-kB	258.3 ± 31.2	2296.2 ± 219.9	800/4000	11.2
GAPDH	443.5 ± 16.1	11,578.4 ± 668.4	5000/4000	3.8
ALB	321.5 ± 30.7	12,096.5 ± 772.1	2000/4000	2.7
B-ACT	297.8 ± 13.6	26,604.0 ± 1024.3	2000/4000	1.1
Akt	358.0 ± 47.6	2325.1 ± 52.5	800/4000	15.4
p-Akt	154.4 ± 10.7	767.4 ± 22.0	800/4000	20.1
p38	299.7 ± 33.6	11,710.3 ± 767.7	800/4000	2.6
p-p38	216.1 ± 12.2	7477.2 ± 271.0	800/4000	2.9
PPARg	194.2 ± 18.7	4351.9 ± 225.0	800/4000	4.5
p-PPARg	384.0 ± 68.8	21,361.2 ± 2108.0	800/4000	1.8
Erk 1/2	237.9 ± 29.4	3077.8 ± 37.3	800/4000	7.7
p-Erk 1/2	409.0 ± 50.7	9374.8 ± 699.4	800/4000	4.4
AMPK	406.5 ± 31.6	10,232.1 ± 452.5	800/4000	4.0
p-AMPK	289.0 ± 62.8	7343.8 ± 436.9	800/4000	3.9

**Table 2 ijms-24-06564-t002:** Intraassay and interassay imprecision data of the multiplexed protein expression assay. For the intraassay study, GAPDH test was performed (N = 84); for interassay analysis, multiparametric test (2 different proteins) was performed (4–8 independent experiments with 4 technical replicates). Data are given as fluorescence/protein arbitrary units (FPR %) and in case of 10 µM OTA treatment as % of the control.

Intraassay Imprecision	FPR
GAPDH (N = 84)	100 ± 9.08 (mean ± SD)
Interassay Imprecision by Independent Measurements	
GAPDH control (N = 8 × 4)	100 ± 5.94 (mean ± SD)
GAPDH OTA treated (N = 6 × 4)	101.1 ± 7.71 (mean ± SD)
B-ACT (N = 4 × 4)	100 ± 4.74 (mean ± SD)
B-ACT OTA treated (N = 4 × 4)	103 ± 4.84 (mean ± SD)

## Data Availability

The data that support the findings of this study are available from the corresponding author upon reasonable request.

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
