# Peer review of "Multiplexed Fluorescence Plate Reader In Situ Protein Expression Assay in Apoptotic HepG2 Cells"

_ijms, 2023, doi:10.3390/ijms24076564_

Round 1
Reviewer 1 Report
In the present study, the authors have detailed a multiplexed fluorescence assay that enables the simultaneous detection of diverse intracellular proteins. The authors have successfully demonstrated that the aforementioned assay can be completed within a timeframe of 6-7 hours and have showcased its utility by assessing apoptosis-related proteins or phosphoproteins in drug-treated HepG2 cells. While the findings are noteworthy, it is pertinent to address several critical issues prior to considering the manuscript for publication.
Firstly, it would be beneficial to include a schematic illustration or flow chart outlining the methodology in order to improve the clarity of the content and design for readers. Secondly, it would be helpful to provide information on the lowest detection limit for proteins (in pg,m ng or mg/mL). Furthermore, it is advisable that the authors refrain from making any theoretical claims that are not substantiated by experimental evidence. Lastly, in order to ensure the robustness of the results, it is recommended that they be validated using other gold standard techniques.
Reviewer 2 Report
In this study, Jakabfi-Csepregi et.al. developed a multiplex protein quantification method that can be used for the quantification of multiple proteins in 96-well cell culture conditions. Such a method may be useful for molecular research. However, the paper is not well-written and needs to be revised extensively. My comments are given below:
1. Abstract: The authors need to write the main results in the abstract clearly. I could not understand their results by reading the abstract.
2. Introduction:
a) In the introduction, the authors need to write the background clearly. I suggest the authors write the conventional methods of protein quantification, their advantages, and their disadvantages one by one. Like what are the advantages and disadvantages of Western blotting, then what are the advantages and disadvantages of ELISA or Immunostainings, etc? They tried to write the disadvantages of Western blotting that here cell lysis SDS PAGE and various types of blotting procedures, blocking and immune reaction. However, Western blotting works pretty accurately for the quantification of proteins and is very popular among researchers. It is more time-consuming of course than ELISA-type methods. The authors should think about this matter and change the introduction accordingly.
b) There are many methods similar to the methods described by the authors. The authors need to discuss those methods in the introduction.
c) In the introduction section (2nd paragraph), the authors tried to describe the methods they used. However, such a description should be discussed in the Methods section and the result section.
3. Results:
a) Antibody dilution optimization: the authors need to show the results in figure or table form.
b) cell number optimization: the authors wrote that ‘However, less than 25,000 cells in the wells showed a much higher response (FPR) indicating that OTA could exert a stronger effect on the cells (relatively higher OTA/unit cell number)’. In figure 1a, it is shown that the fluorescence protein ratio (FPR) when normalized, FPR was not increased. Here, many questions arise, with what the value was normalized, what is the meaning of fluorescence and protein? The authors need to explain here.
Reviewer 3 Report
This paper does not have a high priority for publication in Int J Mol Sci for the following reasons:
Major concerns
1. The title is not appropriate. The authors mean the title, "In situ protein expression assay using a multiplex fluorescence plate reader in HepG2 cell cultures." And I believe it would be preferable to submit to another journal, such as Analytical Biochemistry.2. It doesn't even fit the title of the special issue.
Minor concerns
3. Add image data to make each figure. Show me the match between PCR and WB.
4. Please add data from other types of cell lines besides HepG2.
Round 2
Reviewer 1 Report
The authors answered my questions and I don't have any further questions.
Reviewer 2 Report
The quality of the manuscript has improved significantly.
Reviewer 3 Report
Thank you for your revision, but the reviewer could not find the requisite defense. The reviewer is requesting responses again as suggested in the first round.
1. If the authors are given an appropriate answer, the manuscript will be rejected.
2. Please make sure to add image data that can support the data presented in each table.
3. The editorial office will give you more than one month to re-experiment with other cell types.